# VColRL: Learn to Solve the Vertex Coloring Problem Using Reinforcement Learning

## Abstract

We present VColRL, a reinforcement learning framework designed to solve the vertex coloring problem (VCP), where the objective is to assign colors to the vertices of a graph with the minimum number of colors, such that no two adjacent vertices share the same color. The framework is built on a novel Markov Decision Process (MDP) configuration to effectively capture the dynamics of the VCP, developed after evaluating various MDP configurations. Our experimental results demonstrate that VColRL achieves competitive performance in terms of using fewer colors as compared to advanced mathematical solvers and other metaheuristic approaches while being significantly faster. Additionally, our results show that VColRL generalizes well across different types of graphs.

## 1 Introduction

Combinatorial optimization is a branch of mathematical optimization that focuses on identifying the best solution from a finite collection of possibilities. Common examples of combinatorial optimization problems include the traveling salesman problem (TSP, (Voigt, 1831)), the maximum independent set problem (MIS, (Miller & Muller, 1960)), and the vertex coloring problem (VCP, (Birkhoff, 1912)). However, most combinatorial optimization problems are NP-hard, making finding exact solutions in many real-world scenarios impractical.

Over the years, researchers have developed efficient heuristics (e.g. (Knuth, 1997)) to address these problems. However, these heuristics are often problem-specific and require effective application of domain knowledge. In addition to heuristic approaches, researchers have developed advanced optimization solvers such as the Gurobi Optimization Studio (Gurobi Optimization, 2023) and IBM ILOG CPLEX Optimization Studio (IBM, 2024), which excel at finding exact solutions for complex problems. These solvers utilize sophisticated mathematical techniques and algorithms, such as branch-and-bound (Morrison et al., 2016) and cutting-plane methods (Lee et al., 2015), (Goffin & Vial, 2002), to handle large-scale integer, and mixed-integer programming problems with high precision. Despite their powerful capabilities, the compute and time required for these solvers to find optimal solutions increase substantially as the problem size grows (Luppold et al., 2018).

An alternative approach to tackling complex optimization problems is through data-driven methods. These methods provide a simpler structure and do not require extensive domain expertise. By leveraging Machine Learning, data-driven approaches focus on learning from data and performing straightforward computations, such as matrix multiplications (Burkov, 2019). This simplicity often results in significant time savings, as these techniques can process large datasets and yield solutions more rapidly compared to traditional solvers for bigger problem instances. Over the past decades, the rise of neural networks and advanced ML methods has further enhanced these data-driven techniques, making them more powerful and versatile.

While supervised learning offers one path, obtaining labeled datasets can be challenging. As a result, alternative methods are needed. Reinforcement learning presents a promising option as it does not rely on labeled data. Instead, it frames the problem as a Markov Decision Process (MDP), allowing the system to learn optimal strategies through iterative interactions with the environment and feedback (Sutton & Barto, 2018), (Ernst & Louette, 2024).

Deep reinforcement learning (DRL), a subset of reinforcement learning, has gained prominence due to its ability to handle high-dimensional state and action spaces using deep neural networks. DRL

combines reinforcement learning with deep learning to model complex environments and discover intricate patterns that are difficult to capture with traditional methods (Lapan, 2020). By leveraging deep neural networks, DRL can learn sophisticated policies and make decisions based on large and complex datasets, further enhancing its effectiveness in solving complex optimization problems. This approach has proven particularly useful in areas such as robotics (Morales et al., 2021), autonomous systems (Kiran et al., 2021), and game playing (Dong et al., 2020), where it can achieve remarkable performance by learning from interactions with the environment.

For graph-related problems, Graph Neural Networks (GNNs, (Scarselli et al., 2008)) provide a significant advancement. GNNs are tailored to work with graph-structured data, capturing vertices' intricate relationships and dependencies. They utilize a message-passing approach to collect information from neighboring vertices and update the representation of each vertex accordingly. This method allows GNNs to effectively model complex graph structures. Such capabilities are particularly beneficial for vertex coloring problems, where a deep understanding of graph topology is crucial for achieving optimal color assignments. Furthermore, integrating GNNs with DRL can enhance their effectiveness by combining learned graph representations with reinforcement learning strategies, thereby improving performance.

**Our Contribution:** This paper introduces VColRL, a novel reinforcement learning framework tailored for solving the VCP using the PPO (Schulman et al., 2017) algorithm in conjunction with the GraphSAGE (Hamilton et al., 2017) architecture. VColRL leverages an episodic MDP that models the dynamics of graph coloring by defining states, actions, transitions, and rewards. Central to our approach is a novel reward strategy that treats the color set as ordered, penalizing non-contiguous assignments to ensure unique and efficient solutions in terms of color usage. By focusing on minimizing the highest-numbered color used, this strategy improves learning stability and reduces ambiguities in the solution space.

In addition, we incorporate a detailed study of rollback mechanisms for conflict resolution. Unlike prior methods (Ahn et al., 2020) that rely solely on *hard rollback*, we systematically evaluate both *hard* and *soft rollback* methods to experimentally demonstrate that *hard rollback* provides superior performance. Details about these rollbacks are discussed in section 3.1. Our framework also extends the *deferral* action strategy, originally proposed by Ahn et al. (2020) for the Maximum Independent Set (MIS) problem, to address the unique challenges of VCP. By deferring decisions for certain vertices, the framework simplifies subproblems and enhances the agent's ability to minimize color usage. To the best of our knowledge, VColRL is the first framework to incorporate the *deferral* action in the context of VCP.

Through an exhaustive evaluation of configurations combining different rollback strategies, reward mechanisms, and action models, we identify the optimal design for the VCP. Extensive experiments on synthetic graphs and benchmark datasets showcase that VColRL achieves competitive results compared to state-of-the-art solvers and metaheuristic algorithms while maintaining significantly lower computational costs. Furthermore, the framework demonstrates strong generalization across graph types and scalability to larger problem instances, making it a robust solution for the VCP.

## 2 RELATED WORK

Vertex coloring problem is a well-known NP-hard combinatorial optimization problem (Garey & Johnson, 1976). Exact solvers for this problem require exploring an exponentially large solution space to guarantee optimality, which becomes computationally infeasible as the graph size grows (de Lima & Carmo, 2018). Solutions to the vertex coloring problem can be categorized into two main approaches: conventional methods and machine learning-based approaches.

**Conventional Methods:** This category includes approximation algorithms such as DSatur, Welsh-Powell (Aslan & Baykan, 2016), as well as mathematical solvers like Gurobi and CPLEX, which provide exact solutions but at a high computational cost, especially for large graphs. Additionally, metaheuristic approaches such as Simulated Annealing, Genetic Algorithms, and Tabu Search are commonly used for solving the VCP (Mostafaie et al., 2020), (Dokeroglu & Sevinc, 2021). They are capable of generating high-quality and near-optimal results.

**Machine Learning-Based Approaches:** Several studies have used supervised learning for solving the vertex coloring problem. Das et al. (2019) have introduced a supervised learning approach where

a deep learning model, using Long Short-Term Memory layers followed by a correction phase, is trained on optimal coloring of graphs to handle invalid color assignments. Over time, researchers have recognized the need for graph-specific architectures to better address the complexities inherent in vertex coloring problems. Lemos et al. (2019) has employed Graph Neural Networks (GNN) (Scarselli et al., 2008) to predict the chromatic number of graphs, leveraging GNNs' ability to capture graph structures. Similarly, Ijaz et al. (2022) has utilized GNNs to solve the vertex coloring problem, focusing on efficiently finding the chromatic number of large graphs.

However, as graph sizes increase, obtaining ground truth solutions becomes impractical, leading to the emergence of reinforcement learning as a promising solution. Huang et al. (2019) has adapted AlphaGo Zero (Silver et al., 2017) with graph embeddings, introducing a novel deep neural network architecture known as FastColorNet for the vertex coloring problem. Cummins & Veras (2024) have highlighted the potential of RL to tackle the vertex coloring problem but also pointed out its limitations without label-invariant representations. They have emphasized the importance of integrating GNNs to enhance RL performance by providing essential structural insights. Gianinazzi et al. (2021) has proposed a greedy combined probabilistic heuristic for the vertex coloring problem that integrates reinforcement learning and an attention mechanism (Vaswani, 2017) for vertex selection. In their framework, they only used a terminal step reward based solely on color count. Similarly, Watkins et al. (2023) introduced ReLCol, a method that combines Q-learning with GNN for feature extraction.

The most closely related work is by Ahn et al. (2020), which applies reinforcement learning to the MIS problem using the *deferral* action strategy. While effective for binary-state problems, their framework does not naturally extend to the VCP, where permutations of color assignments result in multiple equivalent solutions with the same number of colors, creating a non-unique solution space that complicates learning and slows convergence. VColRL addresses these challenges with a novel reward strategy that minimizes the highest-numbered color used, ensuring solution uniqueness, improving learning stability, and accelerating convergence time. Although both works utilize GraphSAGE (Hamilton et al., 2017) architecture, it is not central to VColRL's approach. GraphSAGE serves as one of several possible architectural choices for capturing graph structure and can be replaced with other graph neural network architectures without affecting the core contributions of VColRL, which lie in its MDP design, reward strategies, and exhaustive evaluation of MDP configurations. These elements, combined with the adaptation of the *deferral* action strategy proposed by Ahn et al. (2020) to the multi-state complexities of VCP, set VColRL apart as a scalable and effective solution for graph coloring.

## 3 Framework for Vertex Coloring Problem

We now describe our framework for the VCP. Given a graph $\mathcal{G} = (\mathbb{V}, \mathbb{W})$ with vertex set $\mathbb{V}$ and edge set $\mathbb{W}$, the objective is to assign colors to the vertices of a graph with the minimum number of colors, so that no two adjacent vertices share the same color. In our approach to the VCP, we begin by attempting to color the vertices using as few colors as possible from a set $\mathbb{C} = \{1, 2, \ldots, n\}$, where $n$ is initially assumed to be large enough to cover all vertices. If any vertices cannot be colored due to constraints, they are temporarily left uncolored and addressed separately to complete the solution. The resulting solution can be represented as a vector $\boldsymbol{x} = [x_i : i \in \mathbb{V}] \in (\{0\} \cup \mathbb{C})^{\mathbb{V}}$, where each element $x_i$ either indicates the color assigned to vertex $i$ from the set $\mathbb{C}$ or $x_i = 0$ denotes that vertex $i$ has not been assigned any color.

### 3.1 Markov Decision Process for the VCP

We model the VCP as a finite MDP, which terminates when either all vertices are colored or the time limit (episode length) is reached.

The key components of the MDP are:

**States:** States represent the current configuration of the system. A state is represented by a vertex-state vector $\boldsymbol{s} = [s_i : i \in \mathbb{V}] \in (\{*, 0\} \cup \mathbb{C})^{|\mathbb{V}|}$, where $s_i \in \mathbb{C}$ denotes the color assigned to vertex $i$, and $s_i = *$ indicates that the vertex is undecided, meaning it is yet to be colored. Initially, all vertices are undecided ($s_i = *$ for all $i \in \mathbb{V}$). The process ends when no undecided vertices remain

or the time limit is reached. After termination, any vertex $i$ that remains undecided will be referred to as uncolored and will be assigned $s_i = 0$.

**Actions:** Actions represent the decisions the agent makes for the undecided vertices. Let $\mathbb{V}_*$ denote the set of undecided vertices. We consider two models for action:

- *Model with Deferral:* In this model, the agent can either defer the decision for an undecided vertex or assign it a color. This is represented by $\boldsymbol{a}_* = [a_i : i \in \mathbb{V}_*] \in (\{*\} \cup \mathbb{C})^{|\mathbb{V}_*|}$, allowing the agent to delay coloring certain vertices and focus on smaller parts of the graph first.

- *Model without Deferral:* In this model, the agent must assign a color to each undecided vertex without the option to defer. This is represented by $\boldsymbol{a}_* = [a_i : i \in \mathbb{V}_*] \in \mathbb{C}^{|\mathbb{V}_*|}$.

**Transitions:** Transitions define how the system moves from one state to another after an action is taken. The transition from state $\boldsymbol{s} \to \boldsymbol{s}'$ for action $\boldsymbol{a}_*$ occurs in two phases: the update phase and the clean-up phase.

*Update Phase:* The action $\boldsymbol{a}_*$, determined by the policy for the undecided vertices $\mathbb{V}_*$, is applied to create an intermediate state $\hat{\boldsymbol{s}}$. Specifically, $\hat{s}_i = a_i$ if $i \in \mathbb{V}_*$, and $\hat{s}_i = s_i$ otherwise.

*Clean-up Phase:* The clean-up phase ensures the resulting state $\boldsymbol{s}'$ is conflict-free. For each pair of conflicting vertices (i.e., vertices assigned the same color), we consider two rollback models:

- *Hard Rollback Model:* Both conflicting vertices are reset to the undecided state, providing greater flexibility to revisit and resolve conflicts, though this may require more steps.

- *Soft Rollback Model:* Only the vertices involved in the latest action are reset to the undecided state, leaving previous assignments unchanged. This approach resolves conflicts more quickly but offers fewer opportunities to adjust earlier assignments.

**Rewards:** The immediate reward for a transition $\boldsymbol{s} \to \boldsymbol{s}'$ is a weighted combination of two terms, namely vertex satisfaction reward and color usage penalty. Vertex satisfaction reward, representing the increase in the number of assigned vertices, is measured as $Sat(\boldsymbol{s}') - Sat(\boldsymbol{s})$, where $Sat(\boldsymbol{s}) = \sum_{i \in \mathbb{V}} \mathbb{I}(s_i \in \mathbb{C})$, and $\mathbb{I}$ is the indicator function.

For the color usage penalty, we consider two models, both of which serve the same purpose of minimizing color usage but represent the penalty in different ways:

- *Max-color strategy:* Penalizes based on the highest-numbered color assigned from $\mathbb{C}$, defined as:
$$UB(\boldsymbol{s}) = \begin{cases} 0, & \text{if } s_i = *, \forall i \in \mathbb{V}, \\ \max\{s_i \mid s_i \in \mathbb{C}; \forall i \in \mathbb{V}\}, & \text{otherwise.} \end{cases}$$

- *Color-count strategy:* Penalizes based on the total number of distinct colors used, defined as
$$Count(\boldsymbol{s}) = \begin{cases} 0, & \text{if } s_i = *, \forall i \in \mathbb{V}, \\ |\{s_i \mid s_i \in \mathbb{C}; \forall i \in \mathbb{V}\}|, & \text{otherwise.} \end{cases}$$

The *max-color* strategy encourages contiguous color assignment starting from the lowest-numbered color in the set $\mathbb{C}$ since the penalty is determined solely by the highest-numbered color assigned. In contrast, the *color-count* strategy penalizes based on the total number of distinct colors used, regardless of their positions.

For a transition $\boldsymbol{s} \to \boldsymbol{s}'$, the immediate reward using the *max-color* strategy is given by $r_{\boldsymbol{s} \to \boldsymbol{s}'} = w_1 \cdot [Sat(\boldsymbol{s}') - Sat(\boldsymbol{s})] + w_2 \cdot [UB(\boldsymbol{s}) - UB(\boldsymbol{s}')]$, whereas for the *color-count* strategy, the reward is calculated similarly, with the $UB$ function replaced by the $Count$ function.

Since our objective is to ensure that all vertices are colored, we should prioritize increasing vertex satisfaction reward over reducing the color usage penalty. This can be achieved by ensuring that the weights satisfy $w_1 > w_2$. This choice is crucial because, for example, if $w_1$ and $w_2$ are equal, a situation could arise where coloring one more vertex by using one new color results in a zero reward, effectively making it equivalent to taking no action at all.

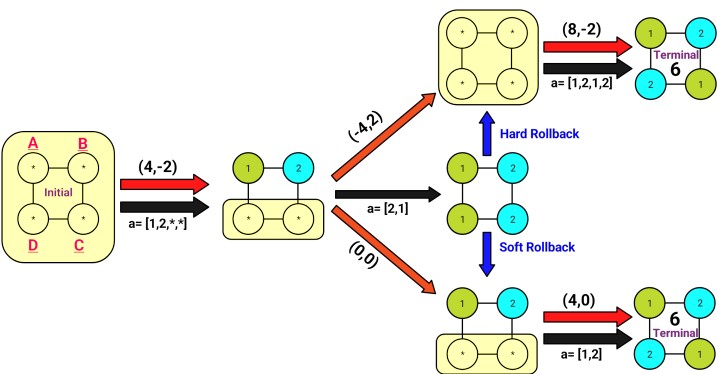

*Figure 1:* **Illustration of the MDP for the VCP** with $w_1 = 2$ and $w_2 = 1$. *The red arrow represents the final transition, the black arrow represents the update phase, and the blue arrow represents the cleanup phase. The sum of elements in the tuples (vertex satisfaction reward, color usage penalty) denotes the immediate reward for a transition, while the numerical value in the terminal state represents the total returns, calculated by summing the rewards of all transitions in the episode. The vertices highlighted in yellow indicate the subgraphs where actions are taken, with the action vector* **a** *ordered by vertices A, B, C, and D, maintaining the same vertex order within the subgraphs.*

**Illustrative example of VColRL's underlying MDP:** Figure 1 illustrates the vertex coloring process for a graph with four vertices labeled A, B, C, and D. The initial state vector is $[*, *, *, *]$, indicating that no vertices have been assigned colors yet. An action $[1, 2, *, *]$ is taken, assigning colors 1 and 2 to vertices A and B, respectively, resulting in the next state $[1, 2, *, *]$. This transition yields a vertex satisfaction reward of $w_1 \times 2 = 4$, where $w_1 = 2$ is the weight for vertex satisfaction, and 2 vertices are satisfied. The color usage penalty is $w_2 \times -2 = -2$, where $w_2 = 1$ is the weight for color usage, and 2 colors are used. This results in the tuple $(4, -2)$, where 4 corresponds to the weighted vertex satisfaction reward, and -2 corresponds to the weighted color usage penalty. The total transition reward for this step is obtained by summing these values, yielding a reward of 2. Subsequently, action $[2, 1]$ is applied to the undecided vertices C and D, resulting in an intermediate state of $[1, 2, 2, 1]$. This configuration leads to a conflict, requiring the application of rollback strategies during the clean-up phase. Depending on the specific rollback strategy used, the transition will proceed as follows:

In the *hard rollback* strategy, all vertices are reverted to the undecided state, resulting in a state vector $[*, *, *, *]$ and a final tuple of $(-4, 2)$. The reward and penalty are calculated based on reverting all assignments. The action $[1, 2, 1, 2]$ is then applied leading to the final state $[1, 2, 1, 2]$ with a reward tuple of $(8, -2)$, giving a total episode return (i.e., the cumulative reward in that episode) of 6.

In contrast, the *soft rollback* strategy only reverts the newly assigned vertices, resulting in the state $[1, 2, *, *]$ and a tuple of $(0, 0)$, indicating no additional reward or penalty from the reverted assignments. The action $[1, 2]$ is then applied to the remaining undecided vertices C and D, resulting in the final state $[1, 2, 1, 2]$ with a reward tuple of $(4, 0)$, giving a total return of 6 for the episode.

## 3.2 TRAINING WITH PROXIMAL POLICY OPTIMIZATION FOLLOWING GRAPHSAGE ARCHITECTURE

We use the Proximal Policy Optimization algorithm (PPO, (Schulman et al., 2017)) to train the agent to solve the VCP problem. The objective for the actor is expressed as:

$$\mathcal{L}_{\text{actor}}(\theta) = \mathbb{E}_t \left[ \min \left( r_t(\theta) \hat{A}_t, \text{clip} \left( r_t(\theta), 1 - \epsilon, 1 + \epsilon \right) \hat{A}_t \right) \right]$$

where the subscript $t$ denotes the time within the episode (trajectory), $r_t(\theta) = \frac{\pi(a_t|s_t;\theta_{\text{new}})}{\pi(a_t|s_t;\theta_{\text{old}})}$ is the probability ratio of the new policy to the old policy at time $t$, $\epsilon$ is a hyperparameter that defines the clipping range to prevent large policy updates, $\hat{A}_t$ is the advantage which represents the difference

between the actual return and the estimated value of the state at time $t$. In addition, PPO also has critic loss, and we add entropy regularization to the total objective to encourage exploration. For a detailed explanation of PPO equipped with entropy regularization, refer to Appendix A.3.

We use the GraphSAGE neural network architecture (Hamilton et al., 2017) for the policy and value networks. As discussed in 3.1, our approach focuses on assigning colors to the undecided vertices of the graph by considering their subgraph. Unlike works such as LwD (Ahn et al., 2020), where the determined part of the graph is unaffected by new assignments, in the VCP, new assignments can lead to conflicts with vertices that were colored in previous steps. Therefore, we design a vertex feature that effectively conveys all the necessary information to the subgraph, enabling it to identify which color assignments could lead to conflicts. Consequently, our vertex feature vector is of the length of $1 + |\mathbb{C}|$, where $|\mathbb{C}|$ represents the number of colors in the set $\mathbb{C}$. The first element of the feature vector is set to 1 so that after feature aggregation in the GNN (Appendix A.2), it reflects the vertex's degree. The remaining $|\mathbb{C}|$ elements indicate the color usage status by the vertex's neighbors: 1 if none use the corresponding color, and -1 if one or more do.

For details about the actor and critic networks, model training, and hyperparameters, refer to Appendix A.2 and B.1.

## 3.3 Addressing Incomplete Solutions in the VColRL Framework

The model is designed to output a color assignment for each vertex at each state, requiring the learning of a probability distribution over a fixed set of colors. We set this color set length to 15. VColRL begins with an initial set of 15 colors. An incomplete solution may occur either when the time limit is reached before all vertices are colored, resulting in fewer than 15 colors being used, or when all 15 colors are utilized but some vertices remain uncolored. In either case, the subgraph of uncolored vertices is extracted, and the coloring process is restarted on the subgraph. This iterative process continues until all vertices are successfully colored. The final color count is obtained by summing the total number of colors used across all iterations.

## 4 Performance Evaluation of VColRL

In this section, we outline the experiments conducted to assess the performance of our proposed method for solving the VCP. For our experiments, we use an NVIDIA GeForce RTX 4090 GPU for model training, with a 12th Gen Intel® Core™ i7-12700 processor featuring 20 cores. To assess our model's performance, we first test it on a diverse set of random synthetic graph types, including Erdős-Rényi (ER, (Erdos et al., 1960)), Barabási-Albert (BA, (Albert & Barabási, 2002)), and Watts-Strogatz (WS, (Watts & Strogatz, 1998)) graphs. Following this, we evaluate our algorithm on the DIMACS and COLOR02 benchmark datasets (Trick, 2002-2004). We train the models using a dataset of 15,000 Erdős–Rényi (ER) graphs with 50 to 100 vertices and an edge probability of 0.15. This data set is divided into two parts: the first 14,000 graphs are used for training, while the last 1,000 graphs are used for validation with a time limit (episode length) value of 32.

## 4.1 Baselines

For our experiments, we use four baselines to compare the performance of our approach. The first baseline is a straightforward **greedy coloring** approach that colors the graph vertices in a sequential manner, ensuring that no two adjacent vertices share the same color. The second baseline is **TabucolMin**, built on top of the Tabucol (Hertz & Werra, 1987) algorithm, which incorporates an additional minimization step to refine the coloring. The third baseline is **VColMIS**, which can be described as follows: given an input graph, we first find the MIS and color it with color '1', then find the MIS on the remaining subgraph and color it with color '2', and so on. For solving the MIS problem, we use the approach discussed in Ahn et al. (2020). Finally, the fourth baseline is the **Gurobi 11** solver (Gurobi Optimization, 2023). The details of these baselines are provided in Appendix B.3.

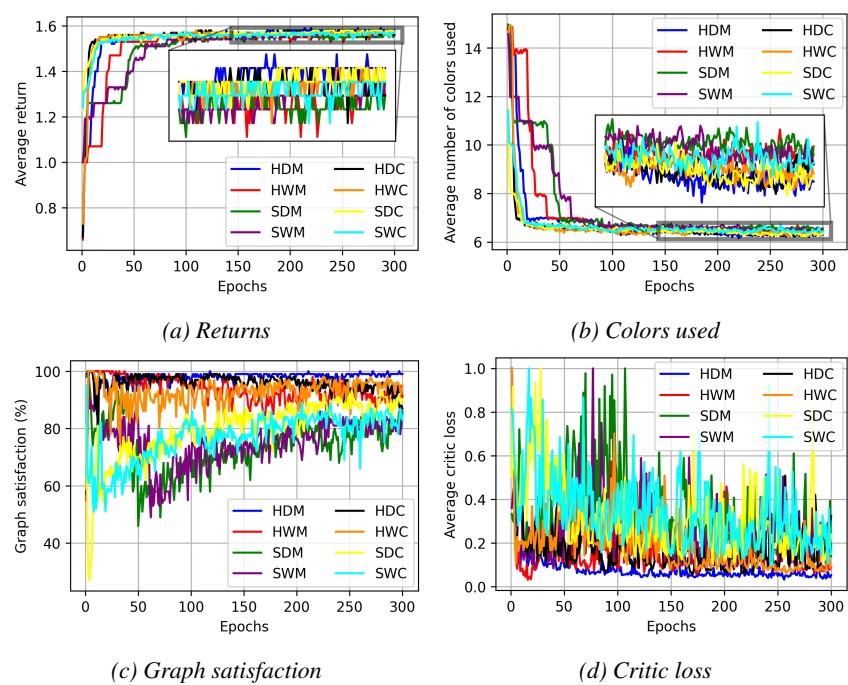

*(a) Returns*        *(b) Colors used*

*(c) Graph satisfaction*        *(d) Critic loss*

Figure 2: **Performance of VColRL with different MDP configurations:** *The strategies used in our framework are represented by the following abbreviations: **H** denotes hard rollback, **S** denotes soft rollback, **D** denotes with deferral action, **W** denotes without deferral action, **M** denotes max-color reward strategy, and **C** denotes color-count reward strategy, e.g., **HDM** denotes hard rollback with deferral action and max-color reward strategy.*

## 4.2 COMPARISON OF VARIOUS MDP CONFIGURATIONS

We first analyze all combinations of rollback models (*soft rollback* and *hard rollback*), action types (*with deferral* and *without deferral*), and reward strategies (*max-color* and *color-count*), resulting in a total of 8 configurations. This comprehensive analysis helps us determine the best MDP configuration.

Figure 2a shows the expected return over (sum of rewards) epochs. Figure 2b illustrates the average number of colors used, reflecting how color usage evolves during training. Figure 2c presents the graph satisfaction percentage, indicating the proportion of validation graphs where all vertices are satisfied. Additionally, Figure 2d shows the average critic losses, normalized independently. This figure is intended to demonstrate the relative stability of variants' critic loss during training, but the relative values may not reflect the true loss levels since they are normalized independently. Together, these figures offer a comprehensive view of the agent's decision-making process, highlighting its impact on color efficiency, graph satisfaction, and critic loss stability.

Figure 2a demonstrates that returns increase for all MDP configurations, indicating that the agent effectively learns to fulfill its objective. Similarly, Figure 2b shows a consistent decrease in the number of colors used across epochs for all configurations, reflecting the agent's ability to minimize color usage. Among these, the HDM (*hard rollback with deferral action and max-color reward strategy*) emerges as the best-performing configuration, achieving higher returns and using fewer colors on average. This is particularly evident between epochs 150–300, where the blue line representing HDM stands out by being above others in returns and below others in average colors used. In Figure 2c, all *hard rollback* configurations initially start with lower graph satisfaction values, gradually increase to 100%, but differ in their post-convergence behavior. While HDM maintains full satisfaction, demonstrating stability, other configurations begin to deteriorate, resulting in instability. In contrast, the *soft rollback* configurations start with high graph satisfaction (near 100%), experience significant drops, and then oscillate before recovering to approximately 90% in 300 epochs, implying slower convergence and longer training times. Not maintaining graph satisfaction indicates that

the agent fails to color all vertices within the allotted time limit. Despite this, Figure 2a shows that returns remain relatively stable for such configurations, suggesting that only a few vertices per graph are left uncolored. However, HDM consistently achieves full graph satisfaction, demonstrating its ability to color graphs more efficiently within less time.

HDM emerges as the best configuration due to several key factors. First, the *hard rollback* mechanism in HDM reverts both conflicting vertices to a deferred state, unlike *soft rollback*, which only reverts the newly assigned vertex. Since reinforcement learning relies on trial and error, *hard rollback* offers greater flexibility, enabling the model to explore better solutions more effectively. Additionally, the *with deferral* action strategy allows the agent to break the problem into simpler subproblems by postponing some decisions to later timesteps. This mirrors human problem-solving, where certain vertices are addressed first to simplify the decision process in subsequent stages. Conversely, in the *without deferral* action strategy, the agent is forced to make decisions immediately, regardless of its confidence, which hinders its performance. Lastly, the *max-color* reward strategy treats the color set as ordered, encouraging the agent to assign colors sequentially from the first color up to $k$, ensuring the solution is unique in terms of the colors used. This contrasts with the *color-count* strategy, where multiple optimal solutions can exist. For instance, if three colors are needed to color a graph, under the color count strategy, both color assignments '1, 2, 3' and '1, 2, 4' will yield a color usage penalty of 3. However, under the *max-color* reward strategy, the former will yield a color usage penalty of 3, while the latter will incur a penalty of 4, ensuring a unique and contiguous solution in terms of used colors in the trained model thereby reducing randomness in the learning process by providing an organized and effective approach for the VCP.

In summary, HDM is an MDP configuration developed by combining our novel *max-color* reward strategy with the *deferral* action strategy proposed by Ahn et al. (2020). This configuration outperforms others in terms of training time and graph coloring efficiency, making it the optimal choice. Figure 2d supports this conclusion, as the critic loss trajectory for HDM demonstrates the highest stability among all.

### 4.3 PERFORMANCE ACROSS DIFFERENT GRAPHS

We now use the best-performing MDP configuration to train VColRL. Further training details can be found in Appendix B.1. Additionally, we train VColMIS on the same dataset using the hyperparameters suggested by Ahn et al. (2020). VColRL is evaluated on a diverse set of graph types and sizes, including ER (Erdos et al., 1960), BA (Albert & Barabási, 2002), WS (Watts & Strogatz, 1998), and graphs from the DIMACS and COLOR02 benchmark datasets (Trick, 2002-2004). Since this is a highly stochastic framework, we evaluate 500 samples per graph. To do this, we combine 500 disconnected instances of each graph into a single graph, which is then fed into the model. After obtaining the output, we report the best solution. The execution time reported for VColRL in this section includes all of these processes.

*Table 1: **Performance comparison across different synthetic graph types and vertex ranges:** The evaluation is conducted on 500 graphs for each of three ranges across the Erdős-Rényi (ER), Barabási-Albert (BA), and Watts-Strogatz (WS) graphs. For each graph, Gurobi's execution time is limited to 10 seconds. Each entry in the table has two values: the first one represents the average number of colors used in those graphs, whereas the second value denotes the average execution time in seconds. The objectives of best-performing methods in terms of minimizing color usage are boldfaced. In ER graphs, $p$ is the probability of edge creation; in BA, $n$ is the number of edges to attach from a new vertex to existing vertices; whereas in WS, $n$ is the number of nearest neighbors to which each vertex is joined within a ring topology, and $p$ is the probability of rewiring each edge.*

| Graph Type | Vertex Range | Greedy | TabucolMin | VColMIS | Gurobi | VColRL |
|---|---|---|---|---|---|---|
| ER ($p$=0.15) | 50-100 | 7.574, $6e^{-5}$ | **5.266**, 4.98 | 7.196, 0.56 | 5.320, 4.87 | 5.452, 0.89 |
| ER ($p$=0.125) | 100-150 | 9.392, $2e^{-4}$ | **6.240**, 12.21 | 8.694, 0.72 | 6.65, 10.22 | 6.544, 1.95 |
| ER ($p$=0.10) | 150-200 | 10.066, $1e^{-4}$ | **6.556**, 14.47 | 9.292, 0.73 | 7.288, 10.27 | 7.144, 1.43 |
| BA ($n$=6) | 50-100 | 6.666, $9e^{-5}$ | 6.072, 4.02 | 9.008, 0.62 | **5.986**, 2.03 | 5.998, 1.02 |
| BA ($n$=6) | 100-150 | 6.846, $1e^{-4}$ | 6.254, 5.55 | 9.886, 0.76 | **6.028**, 5.11 | 6.058, 1.70 |
| BA ($n$=6) | 150-200 | 6.916, $1e^{-4}$ | 6.384, 5.84 | 10.358, 0.81 | **6.045**, 8.04 | 6.180, 2.77 |
| WS ($n$=15, $p$=0.30) | 50-100 | 8.778, $1e^{-4}$ | **6.574**, 7.63 | 8.604, 0.67 | 6.838, 7.52 | 6.906, 1.70 |
| WS ($n$=15, $p$=0.30) | 100-150 | 8.902, $2e^{-4}$ | **6.466**, 10.49 | 8.726, 0.75 | 7.241, 10.17 | 7.016, 2.46 |
| WS ($n$=15, $p$=0.30) | 150-200 | 8.976, $2e^{-4}$ | **6.496**, 10.83 | 8.824, 0.86 | 7.763, 11.14 | 7.028, 3.73 |

*Table 2: **Performance comparison across the DIMACS and COLOR02 benchmark dataset:** Gurobi's time limit is set to 900 seconds. Each entry in the table has two values: the first one represents the number of colors used, whereas the second value represents the execution time in seconds. The best-performing algorithms in terms of using the least number of colors are boldfaced for each graph.*

| Graph Type | Vertices, Edges | Greedy | TabucolMin | VColMIS | Gurobi | VColRL |
|---|---|---|---|---|---|---|
| ash331GPIA | 662, 4181 | $10, 7e^{-4}$ | **4**, 111.02 | 6, 0.49 | **4**, 312.28 | 5, 5.35 |
| ash608GPIA | 1216, 7844 | $9, 1e^{-3}$ | **4**, 254.72 | 6, 0.54 | 8, 900.00 | 5, 11.53 |
| ash958GPIA | 1916, 12506 | $10, 2e^{-3}$ | **4**, 555.93 | 6, 0.56 | 9, 900.00 | 5, 19.37 |
| DSJC125.1 | 125 , 736 | $8, 1e^{-4}$ | **5**, 12.58 | 8, 0.50 | **5**, 46.85 | 6, 1.12 |
| DSJC125.5 | 125, 3891 | $26, 2e^{-4}$ | **18**, 19.32 | 25, 1.52 | 19, 900.00 | 33, 19.59 |
| DSJC125.9 | 125, 6961 | $56, 2e^{-4}$ | **44**, 30.51 | 54, 2.92 | 45, 900.00 | 57, 78.21 |
| DSJC250.1 | 250, 3218 | $13, 3e^{-4}$ | **8**, 46.51 | 12, 0.80 | 9, 900.00 | 12, 16.07 |
| DSJC250.5 | 250, 15688 | $43, 9e^{-4}$ | **29**, 140.17 | 42, 6.55 | 41, 900.00 | 45, 162.55 |
| DSJC250.9 | 250, 27897 | $99, 1e^{-3}$ | **73**, 229.78 | 101, 5.15 | 90, 900.00 | 105, 639.27 |
| 1-FullIns_3 | 30, 100 | $8, 3e^{-5}$ | **4**, 1.06 | **4**, 0.22 | **4**, 0.30 | **4**, 0.35 |
| 1-FullIns_4 | 99 , 593 | $11, 1e^{-4}$ | **5**, 2.20 | **5**, 0.24 | **5**, 19.27 | **5**, 0.74 |
| 1-FullIns_5 | 282 , 3247 | $14, 5e^{-4}$ | **6**, 8.84 | **6**, 0.37 | **6**, 900.00 | **6**, 2.89 |
| 2-FullIns_3 | 52 , 201 | $10, 5e^{-5}$ | **5**, 0.94 | **5**, 0.16 | **5**, 0.95 | **5**, 0.49 |
| 2-FullIns_4 | 212 , 1621 | $14, 2e^{-4}$ | **6**, 2.31 | **6**, 0.27 | **6**, 170.61 | **6**, 2.40 |
| 2-FullIns_5 | 852 , 12201 | $18, 1e^{-3}$ | **7**, 54.40 | 8, 0.40 | **7**, 900.00 | **7**, 15.40 |
| 3-FullIns_3 | 80 , 346 | $12, 7e^{-5}$ | **6**, 1.04 | **6**, 0.90 | **6**, 2.33 | **6**, 0.68 |
| 3-FullIns_4 | 405 , 3524 | $17, 6e^{-4}$ | **7**, 4.46 | **7**, 0.49 | **7**, 421.08 | **7**, 3.59 |
| 4-FullIns_3 | 114 , 541 | $14, 1e^{-4}$ | **7**, 0.97 | **7**, 0.38 | **7**, 5.66 | **7**, 1.13 |
| 4-FullIns_4 | 690, 6650 | $20, 6e^{-4}$ | **8**, 15.04 | **8**, 0.43 | **8**, 900.00 | **8**, 8.27 |
| 4-FullIns_5 | 4146, 77305 | $26, 9e^{-3}$ | **9**, 1690.21 | 10, 0.71 | **9**, 900.00 | 10, 64.16 |
| 5-FullIns_3 | 154, 792 | $16, 1e^{-4}$ | **8**, 1.14 | **8**, 0.34 | **8**, 6.62 | **8**, 1.69 |
| 1-Insertions_4 | 67 , 232 | $5, 5e^{-5}$ | **5**, 0.62 | **5**, 0.10 | **5**, 44.15 | **5**, 0.42 |
| 1-Insertions_5 | 202 , 1227 | $6, 2e^{-4}$ | **6**, 1.24 | **6**, 0.28 | **6**, 900.00 | **6**, 1.36 |
| 1-Insertions_6 | 607 , 6337 | $7, 9e^{-4}$ | **7**, 12.78 | **7**, 0.21 | **7**, 900.00 | **7**, 6.97 |
| 2-Insertions_3 | 37 , 72 | $4, 2e^{-5}$ | **4**, 0.47 | **4**, 0.18 | **4**, 1.85 | **4**, 0.34 |
| 2-Insertions_4 | 149 , 541 | $5, 1e^{-4}$ | **5**, 0.73 | **5**, 0.21 | **5**, 900.00 | **5**, 0.71 |
| 2-Insertions_5 | 597 , 3936 | $6, 6e^{-4}$ | **6**, 6.31 | 7, 0.23 | **6**, 900.00 | **6**, 4.26 |
| 3-Insertions_3 | 56 , 110 | $4, 4e^{-5}$ | **4**, 0.50 | **4**, 0.13 | **4**, 4.39 | **4**, 0.36 |
| 3-Insertions_4 | 281 , 1046 | $5, 2e^{-4}$ | **5**, 1.25 | **5**, 0.21 | **5**, 900.00 | **5**, 1.39 |
| 3-Insertions_5 | 1406 , 9695 | $6, 1e^{-3}$ | **6**, 35.57 | 7, 0.45 | **6**, 900.00 | **6**, 11.86 |
| 4-Insertions_3 | 79 , 156 | $4, 5e^{-5}$ | **4**, 0.52 | **4**, 0.20 | **4**, 17.44 | **4**, 0.46 |
| 4-Insertions_4 | 475 , 1795 | $5, 3e^{-4}$ | **5**, 2.23 | **5**, 0.21 | **5**, 900.00 | **5**, 2.52 |
| le450_5a | 450 , 5714 | $14, 7e^{-4}$ | **5**, 510.47 | 11, 0.88 | 10, 900.00 | 6, 15.96 |
| le450_5b | 450 , 5734 | $13, 7e^{-4}$ | **5**, 542.17 | 12, 0.99 | 10, 900.00 | 6, 16.36 |
| le450_5c | 450 , 9803 | $17, 1e^{-3}$ | **5**, 863.32 | 9, 1.08 | 8, 900.00 | **5**, 14.72 |
| le450_5d | 450 , 9757 | $18, 1e^{-3}$ | 7, 39.22 | 11, 0.69 | 14, 900.00 | **5**, 13.70 |
| mug88_1 | 88 , 146 | $4, 5e^{-5}$ | **4**, 0.50 | **4**, 0.31 | **4**, 2.25 | **4**, 0.61 |
| mug88_25 | 88 , 146 | $4, 5e^{-5}$ | **4**, 0.52 | **4**, 0.26 | **4**, 1.16 | **4**, 0.64 |
| mug100_1 | 100 , 166 | $4, 6e^{-5}$ | **4**, 0.53 | **4**, 0.29 | **4**, 0.52 | **4**, 0.64 |
| mug100_25 | 100 , 166 | $4, 6e^{-5}$ | **4**, 0.54 | **4**, 0.30 | **4**, 0.53 | **4**, 0.64 |
| myciel3 | 11 , 20 | $4, 1e^{-5}$ | **4**, 0.46 | **4**, 0.14 | **4**, 0.04 | **4**, 0.27 |
| myciel4 | 23 , 71 | $5, 2e^{-5}$ | **5**, 0.63 | **5**, 0.16 | **5**, 0.50 | **5**, 0.32 |
| myciel5 | 47 , 236 | $6, 6e^{-5}$ | **6**, 0.84 | **6**, 0.47 | **6**, 2.76 | **6**, 0.65 |
| myciel6 | 95 , 755 | $7, 1e^{-4}$ | **7**, 1.29 | **7**, 0.31 | **7**, 900.00 | **7**, 1.38 |
| myciel7 | 191, 2360 | $8, 2e^{-3}$ | **8**, 2.87 | **8**, 0.61 | **8**, 900.00 | **8**, 4.12 |
| queen5_5 | 25 , 160 | $8, 4e^{-5}$ | **5**, 5.15 | 7, 0.45 | **5**, 0.10 | **5**, 0.51 |
| queen6_6 | 36 , 290 | $11, 6e^{-5}$ | **7**, 4.91 | 8, 0.51 | **7**, 0.60 | **7**, 1.87 |
| queen7_7 | 49 , 476 | $10, 8e^{-5}$ | **7**, 14.85 | 11, 0.75 | **7**, 0.80 | **7**, 2.94 |
| queen8_8 | 64, 728 | $13, 1e^{-4}$ | **9**, 10.53 | 11, 0.72 | **9**, 22.67 | 10, 4.90 |
| queen9_9 | 81, 1056 | $16, 1e^{-4}$ | **10**, 14.46 | 13, 0.79 | **10**, 900.00 | 12, 7.06 |
| will199GPIA | 701 , 6772 | $11, 9e^{-4}$ | **7**, 19.16 | 10, 0.84 | 8, 900.00 | **7**, 11.13 |

**Performance on Synthetic Random Graphs:** We record the performance of various algorithms on different types of graphs in Table 1, comprising ER (Erdős-Rényi), BA (Barabási-Albert), and WS (Watts-Strogatz) graphs. For each row in this table, we test 500 random graphs. We set a time limit of 10 seconds for Gurobi, as it would otherwise continue searching for optimal solutions for an extended period, making it impractical to evaluate a plethora of graphs within a reasonable timeframe. The parameters of the graphs in each row are set so that VColRL does not take more than 10 seconds.

From Table 1, we observe that VColRL consistently achieves a better objective by using fewer colors than VColMIS and Greedy across all tested graph types and ranges, though it takes more time. In

comparison to Gurobi, VColRL performs better particularly for ER and WS graphs. For BA graphs, the objective of VColRL does not surpass that of Gurobi due to the scale-free nature of these graphs. Additionally, VColRL takes significantly less time than Gurobi across all graph types and ranges. For ER, BA, and WS graphs, VColRL is 5-7×, 2-3×, and 3-4× faster, respectively. In comparison to TabucolMin, VColRL outperforms in terms of minimizing color usage for BA graphs. For ER and WS graphs, VColRL does not surpass TabucolMin; however, the difference in the objective is minimal. Specifically, VColRL, on average, requires only 0.36 more colors for ER graphs and 0.48 more for WS graphs. Although TabucolMin achieves slightly better results for ER and WS graphs, it requires significantly more computation time across all graph types. For ER, BA, and WS graphs, VColRL is 6-10×, 2-4×, and 3-4× faster than Tabucol respectively.

**Performance on DIMACS and COLOR02 Benchmark Dataset:** The performance of various algorithms on a subset of the DIMACS and COLOR02 benchmark dataset (Trick, 2002-2004) is recorded in Table 2. Due to CPU limitations, we only analyze graphs with fewer than 5,000 vertices and less than 100,000 edges. For Gurobi, the time limit is set to 900 seconds, so that it can spend sufficient time to find the solution. In each row, we boldface the best-performing objectives in terms of color usage.

For graphs excluding the DSJC family, VColRL performs better or is at least equivalent to VColMIS and Greedy for all graphs in terms of using fewer colors. In comparison to Gurobi, VColRL uses fewer or equivalent colors for ∼81% of the graphs. Similarly, when compared to TabucolMin, VColRL uses fewer or equivalent colors for ∼73% of the graphs. In terms of execution time, VColRL is significantly faster than Gurobi and TabucolMin for the majority of the graphs.

**Generalization capability and scalability:** From Table 2, we observe that in 37 out of 51 tested graph instances (∼73% of the graphs), VColRL performs better or is at least equivalent to the best-performing baseline in terms of color usage. On the graphs where VColRL underperforms, the average number of extra colors used as compared to the best-performing baseline, excluding the DSJC family, is 1.12, while for the DSJC family, it is 13.5. VColRL is significantly faster than Gurobi and TabuColMin while being slower than VColMIS and Greedy; they offer significantly better solution quality than them. Overall, VColRL provides a balanced trade-off between execution time and solution quality, making it a generalizable and scalable solution for most graphs, with the exception to the DSJC family. The suboptimal performance on DSJC and other graphs can be attributed to the model being trained only on the ER 50-100 dataset with an edge probability of 0.15.

## 5 CONCLUSION

In this paper, we propose VColRL, a deep reinforcement learning framework for solving the VCP. Our method is built on a novel MDP framework, which introduces a reward mechanism designed to minimize the highest-numbered color used from an ordered set. Through extensive experimentation, we show that VColRL outperforms or performs equivalently to the baselines like VColMIS and Greedy in terms of color usage for most of the graph types, and performs competitively with Gurobi and TabucolMin with a significant reduction in execution time. Although VColRL faces challenges with the DSJC family of graphs, its generalization across other graph types demonstrates its potential for graph coloring tasks.

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

# A ADDITIONAL DETAILS ABOUT VCOLRL FRAMEWORK

## A.1 VCOLRL ARCHITECTURE

Figure 3 illustrates the architecture of VColRL, which consists of an agent and an environment. The agent interacts with the environment by exploring different actions and stores key information, including the state, action, reward, and episode termination status, in a replay buffer. The returns and advantages (defined in Appendix A.3)are calculated using data from the replay buffer, which are then utilized by the optimizers as feedback to minimize the total objective function (defined in Appendix A.3).

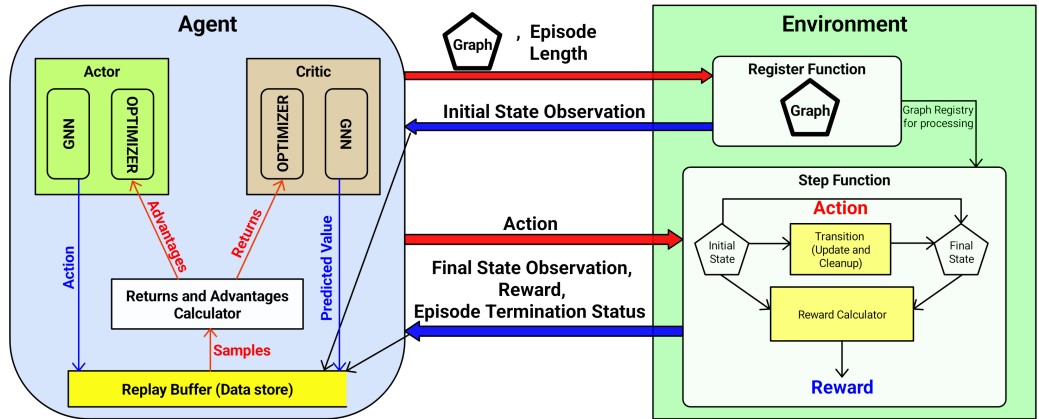

*Figure 3: Architecture of VColRL*

## A.2 GRAPHSAGE ARCHITECTURE

The GNN component in Figure 3 follows the GraphSAGE architecture (Hamilton et al., 2017), illustrated in Figure 4, where the vertex feature vectors are represented using solid arrows, while the flow of information is depicted with regular arrows. It has three modules: Neighborhood Sampling, Feature Aggregation and Concatenation, and Transformation.

In the Neighborhood Sampling module, each vertex samples a few of its neighboring vertices for aggregating their features to be able to capture the information about its neighborhood, but before sampling, the vertex features are normalized by multiplying each feature vector by the inverse square root of the vertices' degree. This normalization helps in stabilizing the learning process. Graph-SAGE typically allows for sampling a subset of neighboring vertices; however, in our approach, we sample the entire neighborhood. This ensures that all relevant vertices are considered for efficient processing, which is essential for accurate feature aggregation.

In the Feature Aggregation and Concatenation module, the features of the sampled vertices are aggregated and normalized again. The aggregated features are then concatenated with the original vertex features, resulting in a feature vector that is twice the length of the input vector. This helps to retain the information about the vertex and its neighborhood, providing a comprehensive representation for further processing.

In the Transformation module, the concatenated and aggregated features (represented as a matrix with the number of rows equals the number of vertices in the graph) are multiplied by a weight matrix, followed by the addition of a bias matrix. Both these matrices are learnable components of our model. An activation function is then applied to produce the final output, enabling the model to capture complex relationships in the data. The final output feature vectors then become the input for the next layer, allowing the model to propagate information through multiple layers.

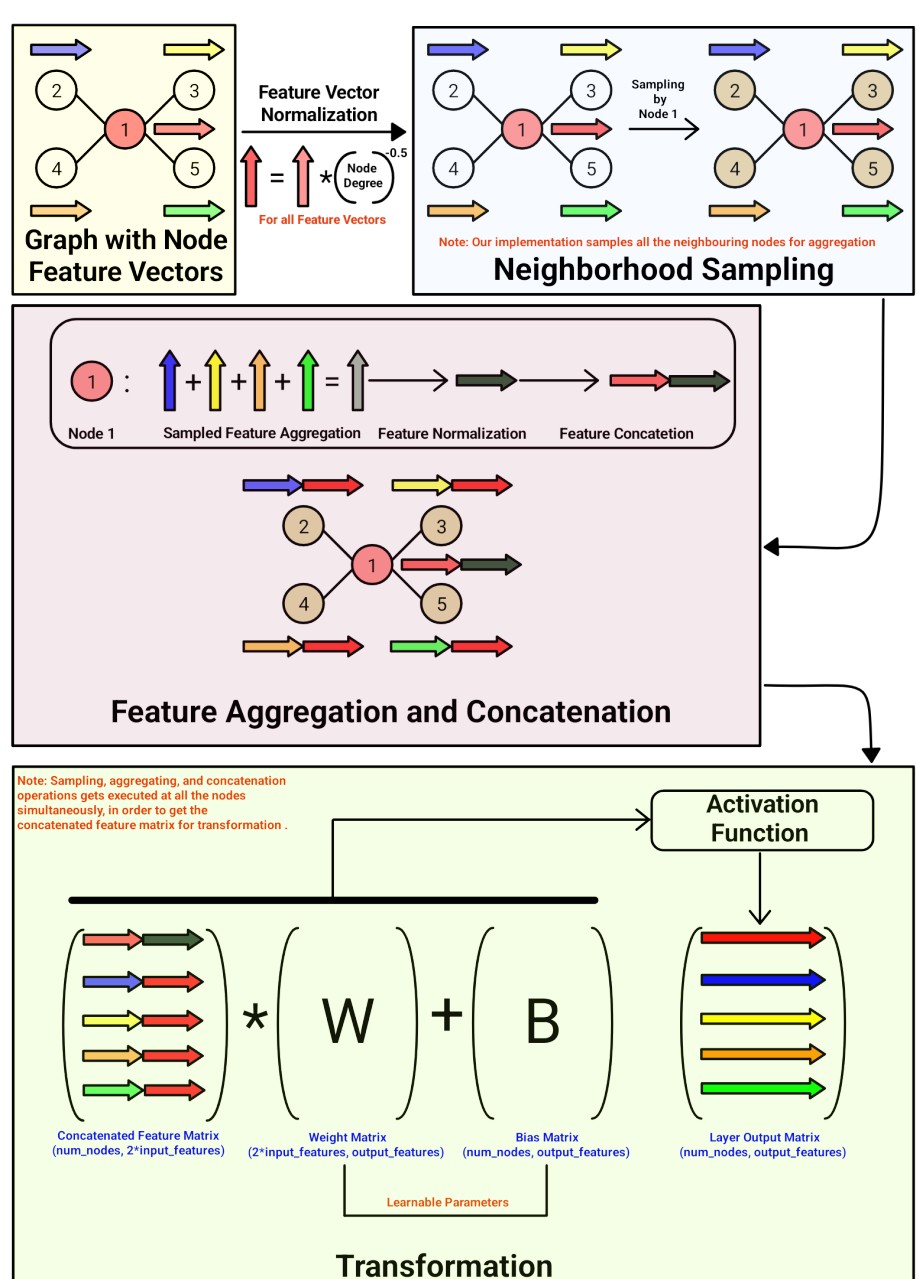

*Figure 4: GraphSAGE architecture*

## A.3 PROXIMAL POLICY OPTIMIZATION WITH ENTROPY REGULARIZATION

The optimizer component in Figure 3 uses the Proximal Policy Optimization algorithm (PPO, (Schulman et al., 2017)) to train the agent to solve the VCP problem. PPO is an actor-critic-based reinforcement learning algorithm that employs two parameterized networks: the policy network $\pi(a_t \mid s_t; \theta)$ and the value network $V(s_t; \omega)$, where $\theta$ and $\omega$ are the parameters of the function estimators. In our case, these are Graph Neural Networks with GraphSAGE (Hamilton et al., 2017) architecture as shown in Figure 4. For more details on the policy and value network architecture, refer to Appendix B.1.

The policy network $\pi(a_t \mid s_t; \theta)$ provides the probabilities of selecting an action $a_t$, given a state $s_t$ at time $t$. The action is determined by sampling from this probability distribution.

On the other hand, the value network $V(s_t; \omega)$ estimates the expected return $\mathbb{E}_\pi[G_t \mid s_t, \omega]$ from the current state $s_t$, where $G_t$ is defined as the discounted sum of future rewards:

$$G_t = \sum_{k=1}^{\infty} \gamma^k r_{t+k}$$

Here, $r_{t+k}$ denotes the reward received at time $t + k$, and $\gamma$ is the discount factor that balances the importance of immediate versus future rewards.

Additionally, PPO uses the advantage function to improve the policy. The advantage function $\hat{A}_t$ is defined as the difference between the actual return and the estimated value of the state:

$$\hat{A}_t = G_t - V(s_t; \omega)$$

The objective function of PPO involves maximizing a surrogate loss function to improve the policy while ensuring stability. The objective function can be expressed as:

$$\mathcal{L}_{\text{actor}}(\theta) = \mathbb{E}_t \left[ \min \left( r_t(\theta)\hat{A}_t, \text{clip}\left( r_t(\theta), 1 - \epsilon, 1 + \epsilon \right) \hat{A}_t \right) \right]$$

where $r_t(\theta) = \frac{\pi(a_t \mid s_t; \theta_{\text{new}})}{\pi(a_t \mid s_t; \theta_{\text{old}})}$ is the probability ratio of the new policy to the old policy, and $\epsilon$ is a hyperparameter that defines the clipping range to prevent large policy updates.

In addition to the policy loss, we have critic loss and entropy regularization to guide the training process. The critic loss is responsible for training the value network $V(s_t; \omega)$ by minimizing the difference between the predicted value from the value network and the actual return from memory. This is done using a squared error loss between the value estimate and the discounted return $G_t$. The critic loss function is given by:

$$\mathcal{L}_{\text{critic}}(\omega) = \mathbb{E}_t \left[ (V(s_t; \omega) - G_t)^2 \right]$$

The entropy regularization term is added to encourage exploration and prevent premature convergence to suboptimal policies. The entropy of the policy $\pi(\boldsymbol{a}_t \mid \boldsymbol{s}_t; \theta)$ measures the uncertainty of the action selection and is defined as:

$$\mathcal{H}(\pi(a_t \mid s_t; \theta)) = -\sum_{a_t} \pi(a_t \mid \boldsymbol{s}_t; \theta) \log \pi(a_t \mid s_t; \theta)$$

For using an optimizer that works by minimizing an objective function, we must adjust the signs of the actor loss and the entropy term accordingly. The actor loss should be negated because we want to maximize the policy objective, while the critic loss remains as it is, as we want to minimize the value estimation error. The entropy term is subtracted to maximize exploration by encouraging higher entropy.

Thus, the total minimization objective function for PPO, combining the actor loss, critic loss, and entropy regularization, is given by:

$$\mathcal{L}_{\text{total}}(\theta, \omega) = -c_1 \mathcal{L}_{\text{actor}}(\theta) + c_2 \mathcal{L}_{\text{critic}}(\omega) - c_3 \mathcal{H}(\pi(a_t \mid s_t; \theta))$$

where $c_1$, $c_2$, and $c_3$, $\in \mathbb{R}^+$ are hyperparameters that control the balance between actor loss, critic loss, and entropy regularization.

In summary, the actor learns to adjust the policy to increase the probability of selecting actions in a state for which the advantage is maximized, thereby promoting actions that yield higher returns. The critic loss helps the agent predict the returns from states, while the entropy regularization encourages the exploration of action space, leading to more robust policies.

## B    ADDITIONAL EXPERIMENTAL DETAILS

### B.1    MODEL TRAINING AND HYPERPARAMETERS

In our experiments, we train the model using a dataset of 15,000 Erdős–Rényi (ER) graphs with 50 to 100 vertices and an edge probability of 0.15. This data set is divided into two parts: the first 14,000 graphs are used for training, while the last 1,000 graphs are used for validation.

Both the actor and critic networks consist of four layers, each with a hidden dimension of 128. ReLU is used as the activation function. The input to both the actor and critic networks is a feature vector described in Section 3.2. We train the model for 300 epochs using the ADAM optimizer (Kingma, 2014). Although this might seem excessive, validation statistics continue to improve even after 200 epochs, i.e., the average number of colors kept decreasing while satisfying all vertices. This is because vertex coloring is a challenging task in reinforcement learning, and extended training helps the model learn more effectively. Figures 2b and 2c demonstrate this improvement in validation metrics during the training period. The best-performing model is determined based on its ability to completely satisfy all validation dataset graphs while utilizing the least number of colors.

The model configurations and hyperparameters are selected through manual tuning and insights from prior works on similar problems. The number of layers is determined empirically by testing various configurations to avoid the over-smoothing issue commonly observed in Graph Neural Networks due to excessive message passing (Chen et al., 2020). After experimentation, we settle on four layers. For the number of neurons per layer, we test values of 16, 32, 64, 128, and 256, finding that increasing the number beyond 128 does not lead to performance improvements. For hyperparameters, we start with values suggested by Ahn et al. (2020) for the MIS problem and then explore nearby ranges. After evaluating various combinations, the final set of hyperparameters available in Table 3 is selected.

*Table 3: Hyperparameters used for training.*

| Hyperparameter | Value |
|---|---|
| Episode Length/ Time Limit | 32 |
| Replay buffer size | 32 |
| Batch size | 32 |
| Batch size for gradient step | 16 |
| Number of gradient steps per update | 4 |
| Actor loss coefficient | 1 |
| Critic loss coefficient | 0.25 |
| Entropy regularization coefficient | 0.01 |
| Learning rate of optimizer | 0.0001 |
| Gradient Norm | 1 |
| Discount factor for PPO | 1 |
| Clip Value for PPO | 0.2 |
| Vertex satisfaction reward weight | 2 |
| Color usage penalty weight | 1 |

### B.2    NORMALIZATION OF REWARDS

Since our immediate reward is the weighted sum of the vertex satisfaction reward and the color usage penalty, we normalize each component by dividing it by a relevant factor before computing the sum. Specifically, the vertex satisfaction reward is divided by the number of vertices, and the color usage penalty is divided by the number of colors in set $\mathbb{C}$. This ensures that both components are scaled appropriately and that neither dominates the reward function. Though it is possible to apply an additional normalization step after combining the two components, we do not apply further normalization in our approach.

### B.3    IMPLEMENTATION OF BASELINES

#### B.3.1    GREEDY COLORING

This baseline assigns colors to vertices in a sequential manner, ensuring that no two adjacent vertices share the same color. The algorithm works by iterating through each vertex and selecting the smallest

---

**Algorithm 1** Greedy Coloring

---

1: $V \leftarrow$ Number of vertices in $Graph$
2: $result \leftarrow$ array of length $V$ initialized with -1
3: $result[0] \leftarrow 0$
4: $available \leftarrow$ array of length $V$ initialized with True
5: $max\_color \leftarrow 0$
6: **for** each vertex $u$ ranging from 1 to $V - 1$ **do**
7:    **for** each neighbor $i$ of vertex $u$ in $Graph$ **do**
8:       **if** $result[i] \neq -1$ **then**
9:          $available[result[i]] \leftarrow False$
10:       **end if**
11:    **end for**
12:    $cr \leftarrow 0$
13:    **while** $cr < V$ **do**
14:       **if** $available[cr] == True$ **then**
15:          $break$
16:       **end if**
17:       $cr \leftarrow cr + 1$
18:    **end while**
19:    $result[u] \leftarrow cr$
20:    **if** $cr > max\_color$ **then**
21:       $max\_color \leftarrow cr$
22:    **end if**
23:    $available \leftarrow$ array of length $V$ initialized with $True$ //resetting available colors
24: **end for**
25: **return** $max\_color +1$

---

available color that has not been assigned to its neighboring vertices. The pseudocode for this is available in Algorithm 1.

### B.3.2 TABUCOLMIN

---

**Algorithm 2** TabucolMin

---

1: $Graph \leftarrow$ Input Graph
2: $N\_Colors \leftarrow 15$
3: $solution \leftarrow 0$
4: $coloring \leftarrow$ list of size $num\_vertices$ in $Graph$, initialized with $-1$
5: $Tabu\_list\_length \leftarrow 6$
6: $max\_iteration \leftarrow 100000$
7: **while** $N\_Colors \geq 1$ **do**
8:    $is\_colorable, coloring \leftarrow$ ***Tabucol***$(Graph, N\_Colors, Tabu\_list\_length, max\_iteration)$
9:    **if** $is\_colorable$ **then**
10:       $solution \leftarrow N\_Colors$
11:       $N\_Colors \leftarrow N\_Colors - 1$
12:    **else if** $solution == 0$ **then**
13:       $N\_Colors \leftarrow N\_Colors + 15$
14:    **else**
15:       **return** $solution, coloring$
16:    **end if**
17: **end while**

---

This algorithm is an iterative metaheuristic designed to minimize the number of colors required to properly color a graph, leveraging the Tabucol algorithm as a subroutine. Hertz & Werra (1987) contains the pseudocode and details about the Tabucol algorithm. TabucolMin begins with an initial color count, typically set to 15, and attempts to color the graph. If the graph can be successfully

colored with the current number of colors, the algorithm reduces the color count by one and retries. Conversely, if the color count cannot be reduced further and no solution has yet been found, the algorithm increases the color count by 15 to expand the search space and retries whereas if the color count cannot be reduced further and a solution has been found, the process stops. Crucially, the reported runtime accounts only for the search space where a solution was successfully found, excluding any computational effort expended in unsuccessful attempts, such as the initial trials with fewer colors. This ensures that the timing reflects only the effective search for the minimal coloring solution. Algorithm 2 presents the pseudocode for this baseline.

### B.3.3 VCOLMIS

---

**Algorithm 3** VColMIS

---

1: $num\_colors \leftarrow 0$
2: $Graph \leftarrow$ Input Graph
3: **while** $Graph$ is not empty **do**
4:    $mis \leftarrow Compute\_MIS\_RL(Graph)$
5:    **if** $mis$ is not empty **then**
6:       $num\_colors \leftarrow num\_colors + 1$
7:    **else**
8:       $num\_colors \leftarrow num\_colors + num\_vertices(Graph)$
9:       **break**
10:   **end if**
11:   $Graph \leftarrow remove\_vertices\_present\_in\_mis\_set(Graph, mis)$
12: **end while**
13: **return** $num\_colors$

---

VColMIS is based on the Maximum Independent Set (MIS) strategy. In this approach, the vertices in the MIS set are colored with the same color. A subgraph is then created from the remaining uncolored vertices, and the process is repeated by finding a new MIS and assigning another color. This process continues iteratively until all vertices in the graph are colored. For the MIS part, we train the RL-based model described by Ahn et al. (2020) on the same dataset used to train our model. This model takes a graph as input and outputs its maximum independent set. We call this model *Compute_MIS_RL*. Algorithm 3 contains the pseudocode for this approach.

### B.3.4 GUROBI 11 SOLVER

Given a color set $\mathbb{C}$ and graph $\mathcal{G} = (\mathbb{V}, \mathbb{W})$, we use the Gurobi 11 Optimizer (Gurobi Optimization, 2023) to solve the following integer linear programming model for the vertex coloring problem.

$$\text{Minimize} \quad \sum_{c=1}^{|\mathbb{C}|} z_c$$

$$\text{subject to} \quad x_{v,c} + x_{u,c} \leq z_c, \quad \forall (u,v) \in \mathbb{W}, \quad \forall c \in \mathbb{C}$$

$$\sum_{c=1}^{|\mathbb{C}|} x_{v,c} = 1, \quad \forall v \in \mathbb{V}$$

$$x_{v,c} \in \{0,1\}, \quad z_c \in \{0,1\}, \quad \forall v \in \mathbb{V}, \quad \forall c \in \mathbb{C}$$

Here, $x_{v,c}$ is a binary variable that indicates whether vertex $v$ is assigned color $c$, and $z_c$ is a binary variable that indicates whether color $c$ is used in the solution.

The algorithm starts with an initial color set size of 15 and iteratively increases the size by 15 if the solver fails to find a feasible solution. The reported runtime includes only the computational effort spent in the search space where a solution is successfully found, excluding any time spent on earlier, unsuccessful attempts with smaller color sets.

