# OpenReview forum: "VColRL: Learn to Solve the Vertex Coloring Problem Using Reinforcement Learning"
_ICLR.cc/2025/Conference — ICLR 2025 Conference Withdrawn Submission_

### Official Review · Reviewer_HWyU · 2024-11-03

**Soundness:** 3
**Presentation:** 3
**Contribution:** 2
**Rating:** 5
**Confidence:** 3

**Summary:**

The authors tackle solving the Vertex Coloring Problem which is a known NP-hard problem using reinforcement learning. They formulate the problem as a Markov-Decision Process whose dynamics capture the graph coloring process. The agent is tasked with taking actions which include assigning colors to a subset of nodes and deferring other nodes to be assigned later. Two reward functions are proposed, max-color and color-count. An agent is trained using PPO where the graph neural network GraphSAGE is used as the function approximator. The experimental results show that the proposed methods outperform existing baselines.

**Strengths:**

- The paper is well-written and easy to follow. The related work section highlights existing gaps and shows how the proposed method differs from existing work. The main contribution is also well documented.
- The suggested modifications to the MDP formulation and the feature selection in previous work are an elegant approach to the vertex coloring problem, for example, the different roll-back strategies and the two reward functions.
- The proposed method outperforms existing baselines in the larger graph sizes.

**Weaknesses:**

- The paper’s contribution, while valuable, may benefit from additional innovation, as it shares similarities with prior work by Ahn et al. which tackles the Maximum Independent Set also using Deep RL and GraphSAGE architecture. Further discussion on how the work is unique and where the overlap lies could strengthen the value of the work.
- There is not a lot of insight into why the method performs the way it does. For example, the results show that the color count reward is not as good as the max-color reward, but no discussion is provided about why that would be the case.
- In the results section, the selected baselines do not include a machine learning-based approach.
-  In the related work the Vertex coloring problem is described as NP-complete. However, according to my understanding, the vertex coloring problem considered is the optimization version which is NP-hard, and not the decision version.

**Questions:**

- In the comparison of various MDP configurations results section, was the color count reward with other settings than hard rollback and deferred actions considered? If not isn't it possible that color count with a different setting outperforms HDM?

---

> ### Author Response · Authors · 2024-11-27
> **reply to reviewer HWyU 1/2**
>
> We sincerely thank the reviewer for the valuable time and effort put into reviewing our work. Several updates have been incorporated into the submission, and these are highlighted in blue in the rebuttal version. Notable updates include the addition of Tabucol as a baseline and the evaluation of all 8 combinations of MDP configurations in Figure 2 based on reward, action, and rollback models. **Due to word constraints, we put this reply in two parts. This is part 1/2 of the first official response to the reviewer.**
>
> ## **Responses to Questions/Weaknesses**
> ### **Weakness 1:**
>
> While both our work and that of Ahn et al.[1] utilize Deep RL and GraphSAGE, we emphasize that the choice of GraphSAGE is incidental and not central to our contributions. It is one of many possible graph neural network architectures that can facilitate learning node representation. The key novelty of our work lies in addressing specific challenges unique to the Vertex Coloring Problem (VCP), which are not present in the MIS problem. For example, the max-color reward strategy introduces a contiguity constraint in the solution space to eliminate ambiguities in optimal solutions, a challenge that does not arise in the binary-state nature of MIS. This innovation and our comprehensive exploration of rollback mechanisms constitute a significant advancement tailored specifically for VCP. The details of our work's main differences and novel contributions are described below.
>
>   **Exploration and Impact of Rollback Mechanisms in VCP:** A key contribution of our work lies in the investigation of different rollback mechanisms for conflict resolution in node assignments. While prior work in the MIS context employed hard rollback mechanisms, these approaches did not fully explore alternative configurations. We address this gap by incorporating and systematically analyzing both hard rollback and soft rollback mechanisms.  In the hard rollback mechanism, both conflicted nodes revert to a deferred state, ensuring no immediate decision is forced when conflicts arise. In contrast, the soft rollback mechanism only reverts the currently assigned node while retaining the prior decisions. Through empirical validation, we demonstrate that the hard rollback mechanism consistently outperforms soft rollback in the VCP context. This highlights the importance of selecting effective rollback strategies for graph coloring and provides valuable problem-specific insights.
>
>    **Max-color reward strategy to ensure uniqueness in the solution**: In the Maximum Independent Set (MIS) problem, states are binary ({0,1}), representing the inclusion or exclusion of vertices. Extending this idea to the Vertex Coloring Problem (VCP) involves treating colors as discrete states. However, this straightforward extension leads to ambiguity in optimal solutions, as it results in multiple equivalent solutions. For example, assignments like {1,2,3} and {1,2,4} are treated as equally optimal since both use the same number of colors. This formulation naturally boils down to the color-count strategy, which is the natural extension of the binary state concept in the MIS problem to the VCP. While the color-count strategy minimizes the number of colors, it lacks a mechanism to enforce contiguity or uniqueness in the solution. One of the main contributions of our work is to address these challenges through the max-color strategy, which introduces the concept of contiguity in the solution space. By encouraging sequential color assignment starting from the smallest color, this strategy eliminates ambiguity and ensures unique optimal solutions. Experimentally, we confirm that the max-color strategy effectively improves learning efficiency and outperforms the color-count strategy in the VCP context.
>
>    **Adaptation of Defer Action Strategy:**
> While the defer action strategy from Ahn et al.[1] was designed for MIS, we adapted it to VCP, focusing on minimizing color usage. To the best of our knowledge, this is the first application of deferred action in the VCP setting, integrated with our max-color reward strategy for enhanced performance.
>
> **We have summarized these points in sections 1 and 2 of the revised version.**
>
> [1] Ahn, Sungsoo, Younggyo Seo, and Jinwoo Shin. "Learning what to defer for maximum independent sets." In International conference on machine learning, pp. 134-144. PMLR, 2020.
>
> **Responses to remaining weaknesses and questions can be found in Part 2/2 of the reply, following this response.**

---

> ### Author Response · Authors · 2024-11-27
> **reply to reviewer HWyU 2/2**
>
> **This is part 2/2 of the first official response to the reviewer**
>
> ## **Responses to Questions/Weaknesses**
> ### **Weakness 2**
>
> HDM emerges as the best configuration due to several key factors. First, the ***hard rollback*** mechanism in HDM reverts both conflicting vertices to a deferred state, unlike *soft rollback*, which only reverts the newly assigned vertex. As RL relies on trial and error, *hard rollback* offers greater flexibility, enabling the model to explore better solutions more effectively.  The ***with deferral*** action strategy allows the agent to break the problem into simpler subproblems by postponing some decisions to later timesteps. This mirrors human problem-solving, where certain vertices are addressed first to simplify the decision process in subsequent stages, whereas, in the *without deferral* action strategy, the agent is forced to make decisions immediately, regardless of its confidence, which hinders its performance. Lastly, the ***max-color*** reward strategy treats the color set as ordered, encouraging the agent to assign colors sequentially from the first color up to k, ensuring the solution is unique in terms of the colors used. This contrasts with the *color-count* strategy, where multiple optimal solutions can exist. For instance, if three colors are needed to color a graph, under the *color-count* strategy, both color assignments ‘1, 2, 3’ and ‘1, 2, 4’ will yield a color usage penalty of 3. However, under the *max-color* reward strategy, the former will yield a color usage penalty of 3, whereas the latter will incur a penalty of 4. *Max-color* reward strategy, therefore, encourages the model to learn unique and contiguous solutions, thereby reducing randomness in the learning process by providing an organized and effective approach for the VCP. This has been added in section 4.2 of the revised version.
>
> ### **Weakness 3**
> Thanks for the suggestion. Based on the reviewer's suggestion, we identified the ReLCol [2] framework to compare with our solution. It uses deep q-learning and graph neural networks to solve the graph coloring problem. However, the code of this paper is not publicly available. Given the short time during the rebuttal phase, we could not implement it on our own to perform a direct comparison. However, the authors in [2] have reported the number of colors used by their solution for some graph types in their paper, which we have compared with our solution. The details are given below:
>
> | Graph            | ReLCol (mean ± std) | VColRL |
> |------------------|---------------------|--------|
> | queen5_5         | 5.2 ± 0.11          | 5      |
> | queen6_6         | 8 ± 0                | 7      |
> | queen7_7         | 9.2 ± 0.16          | 7      |
> | queen8_8         | 11.1 ± 0.14         | 10     |
> | queen9_9         | 12.7 ± 0.25         | 12     |
> | 1-Insertions_4   | 5 ± 0               | 5      |
> | 2-Insertions_4   | 5 ± 0               | 5      |
> | myciel5          | 6 ± 0               | 6      |
> | myciel6          | 7 ± 0               | 7      |
> | myciel7          | 8.2 ± 0.11          | 8      |
> | mug88_1          | 4 ± 0               | 4      |
>
> ### **Weakness 4**
> Thanks for pointing this out. We have updated this in our revised version.
>
> ### **Question 1**
> Thank you for your suggestion. In the revised version, we have updated Figure 2 to include all eight combinations of rollback strategies, action types, and reward strategies, ensuring a comprehensive comparison. This analysis confirms that HDM remains the most efficient method.
>
> [2] Watkins, George, Giovanni Montana, and Juergen Branke. "Generating a graph colouring heuristic with deep q-learning and graph neural networks." In International Conference on Learning and Intelligent Optimization, pp. 491-505. Cham: Springer International Publishing, 2023.

---

### Official Review · Reviewer_PvHX · 2024-11-03

**Soundness:** 3
**Presentation:** 3
**Contribution:** 3
**Rating:** 5
**Confidence:** 3

**Summary:**

The paper presents VColRL, a novel reinforcement learning framework for solving the Vertex Coloring Problem (VCP). The goal of VCP is to color graph vertices with the minimum number of colors so that no two adjacent vertices share the same color. VColRL uses a custom Markov Decision Process (MDP) to model VCP dynamics, optimizing color usage and generalizing well to diverse graph types.

**Strengths:**

1.VColRL outperforms traditional mathematical solvers and baseline methods, particularly for large and dense graphs, by effectively minimizing color usage.
2.By using a unique reward strategy that prioritizes minimizing the highest-numbered color, VColRL efficiently manages color allocation, reducing convergence time and improving performance. This MDP design is a key factor in achieving both optimal color usage and stable results in the vertex coloring process.

**Weaknesses:**

1.While VColRL demonstrates strong performance on large-scale graphs, its complex architecture and computational demands are relatively high. The paper does not provide sufficient detail on its efficiency or scalability in practical applications, which raises concerns about its usability in resource-limited environments.
2.Although VColRL performs well on most tested graphs, it is less effective on certain types, such as the mug family of graphs. The paper lacks a thorough analysis of why VColRL underperforms on these specific graphs, which slightly weakens its claims of general applicability.
3.The paper discusses important design choices, such as rollback and reward strategies, but does not sufficiently explain the rationale behind these choices. Additionally, while hyperparameter settings and the training process are briefly covered, there is limited explanation on how the final parameter combinations were decided, reducing the model’s reproducibility and interpretability.

**Questions:**

1.How does VColRL perform in terms of computational efficiency in real-world applications?
2.What specific factors contribute to VColRL’s limited performance on certain graph types, such as the mug family?
3.What criteria were used to select the final hyperparameter values and model configurations?

---

> ### Author Response · Authors · 2024-11-27
> **reply to reviewer PvHX**
>
> We sincerely thank the reviewer for the valuable time and effort put into reviewing our work. Several updates have been incorporated into the submission, and these are highlighted in blue in the rebuttal version. Notable updates include the addition of Tabucol as a baseline and the evaluation of all 8 combinations of MDP configurations in Figure 2 based on reward, action, and rollback models.
>
> ## **Responses to Questions/Weaknesses**
>
> **Weakness 1 and Question 1:**
> In this paper, we have studied how to use reinforcement learning to solve the VCP in general. The proposed solution is not explicitly designed for resource-limited environments. However, we have extensively compared our algorithm with other approaches regarding scalability and computational efficiency. From our experimentations, we can observe that VColRL colors most graphs very quickly (within seconds). As observed in Tables 1 and 2, VColRL outperforms VColMIS and Greedy in terms of objective, though it takes more time. It competes quite well with Gurobi and Tabucol on the majority of graphs, with significantly reduced execution time. Overall, we believe VColRL offers a fair balance between solution quality and execution time, demonstrating good scalability. This has been reflected in section  4.3 of the revised version.
>
> **Weakness 2 and Question 2:**
> Thanks for the comment. We acknowledge that VColRL performs well on most of the tested graphs but shows limited performance on some graphs. However, this behavior is also true for any machine learning solutions in the literature, where one solution rarely performs well in all scenarios. Therefore, we report the performance of VColRL for a wide variety of graphs.
> Additionally, VColRL's performance depends on numerous factors, such as the number of training epochs, the kind of graphs they are trained on, etc. To address the reviewer's comment, we have increased the model training time from 260 epochs to 300 epochs. This has improved the MUG family graphs' performance while providing competitive performance across most graph types, except for the DSJC family, which is newly added in our revised version. The suboptimal performance on a few graph instances can be attributed to the fact that we trained the model only on one set of graphs ( ER 50-100). These results may be improved by training our model on a diverse set of graph types; however, we could not try this during the rebuttal phase due to lack of time.
>
> **Weakness 3 and Question 3:**
> HDM emerges as the best configuration due to several key factors. First, the ***hard rollback*** mechanism in HDM reverts both conflicting vertices to a deferred state, unlike *soft rollback*, which only reverts the newly assigned vertex. As RL relies on trial and error, *hard rollback* offers greater flexibility, enabling the model to explore better solutions more effectively.  The ***with deferral*** action strategy allows the agent to break the problem into simpler subproblems by postponing some decisions to later timesteps. This mirrors human problem-solving, where certain vertices are addressed first to simplify the decision process in subsequent stages, whereas, in the *without deferral* action strategy, the agent is forced to make decisions immediately, regardless of its confidence, which hinders its performance. Lastly, the ***max-color*** reward strategy treats the color set as ordered, encouraging the agent to assign colors sequentially from the first color up to k, ensuring the solution is unique in terms of the colors used. This contrasts with the *color-count* strategy, where multiple optimal solutions can exist. For instance, if three colors are needed to color a graph, under the *color-count* strategy, both color assignments ‘1, 2, 3’ and ‘1, 2, 4’ will yield a color usage penalty of 3. However, under the *max-color* reward strategy, the former will yield a color usage penalty of 3, whereas the latter will incur a penalty of 4. *Max-color* reward strategy, therefore, encourages the model to learn unique and contiguous solutions, thereby reducing randomness in the learning process by providing an organized and effective approach for the VCP. This has been added in section 4.2 of the revised version.
>
> The model configurations and hyperparameters were selected through manual tuning and insights from prior works on similar problems. For example, the number of layers was determined empirically by testing various configurations to avoid the over-smoothing issue commonly observed in Graph Neural Networks due to excessive message passing. After experimentation, we settled on four layers. For the number of neurons per layer, we tested values of 16, 32, 64, 128, and 256, finding that increasing the number beyond 128 did not lead to performance improvements. After evaluating various combinations, the final set of hyperparameters was selected, as detailed in Table 3. We have discussed these details in Appendix B.1 of the revised version.

---

### Official Review · Reviewer_2gn8 · 2024-11-04

**Soundness:** 3
**Presentation:** 3
**Contribution:** 3
**Rating:** 5
**Confidence:** 5

**Summary:**

This paper proposes a reinforcement learning framework for solving the vertex coloring problem (VCP). The main idea is to use an episodic Markov Decision Process (MDP) to effectively model the dynamics of the coloring process, utilizing both Proximal Policy Optimization (PPO) and GraphSAGE. The writing is clear and well-organized.

**Strengths:**

1. This paper proposes a reinforcement learning framework that integrates the GraphSAGE architecture for the VCP.
2. A novel reward strategy specifically designed for the VCP is introduced.
3. Extensive experiments demonstrate the effectiveness of the proposed algorithm in comparison to state-of-the-art baselines.

**Weaknesses:**

1. Please clarify the novel features introduced in the MDP formulation, and how these differ from or build upon existing approaches like the defer action strategy..
2. Gurobi should be set to a consistent execution time across different benchmarks, and it would be beneficial to provide a longer execution time result as a reference for the optimal solution.
3. Given that the proposed algorithm employs rollback mechanisms similar to iterative search, it would be appropriate to include an iterative search heuristic (i.e., tabuCol) as a comparison.

**Questions:**

1. How does the proposed algorithm decide when to activate the hard-rollback model versus the soft-rollback model?
2. Is the initial number of colors for each instance incremented starting from zero or from a specific given value?
3. The benchmarks in the COLOR02 table can include both DIMACS and COLOR02 instances. The benchmark instances could be more comprehensive, such as including DSJC125.5, DSJC125.9, DSJC250.1，and others.

---

> ### Author Response · Authors · 2024-11-27
> **Reply to Reviewer 2gn8**
>
> We sincerely thank the reviewer for the valuable time and effort put into reviewing our work. Several updates have been incorporated into the submission, and these are highlighted in blue in the rebuttal version. Notable updates include the addition of Tabucol as a baseline and the evaluation of all eight combinations of MDP configurations in Figure 2 based on reward, action type, and rollback strategies.
>
> ## **Responses to Weaknesses**
> 1. Thank you for the feedback. We take this opportunity to clarify the novel features of our MDP formulation and how they build upon existing approaches, such as the deferral action strategy proposed by Ahn et al. [1].  As detailed below, our work introduces several key innovations to extend reinforcement learning techniques to the Vertex Coloring Problem (VCP).
>
>    **Max-Color Reward Strategy**:  A core contribution is the max-color reward strategy, which introduces a contiguity constraint in the solution space to address ambiguities in optimal solutions. While the Maximum Independent Set (MIS) problem uses binary states ({0,1}) to represent the inclusion or exclusion of vertices, extending this to the VCP involves treating colors as discrete states. However, this straightforward extension, reflected in the color-count strategy, leads to multiple equivalent solutions (e.g., {1,2,3} and {1,2,4} both use the same number of colors).  The max-color strategy eliminates this ambiguity by enforcing sequential color assignment starting from the smallest color, ensuring unique optimal solutions in terms of color usage. This not only improves solution quality but also enhances learning efficiency. Experimentally, we confirm the superiority of the max-color strategy over the color-count strategy.
>
>    **Exploration of Rollback Mechanisms**:   Another key contribution of our work lies in the investigation of different rollback mechanisms for conflict resolution in node assignments. While prior work like Ahn et al. [1]  employed hard rollback mechanisms, these approaches did not fully explore alternative configurations. We address this gap by incorporating and systematically analyzing both hard rollback and soft rollback mechanisms.  In the hard rollback mechanism, both conflicted nodes revert to a deferred state, ensuring no immediate decision is forced when conflicts arise. In contrast, the soft rollback mechanism only reverts the currently assigned node while retaining the prior decisions. Through empirical validation, we demonstrate that the hard rollback mechanism consistently outperforms soft rollback in the VCP context. This highlights the importance of selecting effective rollback strategies for graph coloring and provides valuable problem-specific insights.
>
>    **Adaptation of the Deferral Action Strategy to VCP**:  While Ahn et al. [1]. introduced the defer action strategy for MIS, we adapt and refine it for the VCP to address unique challenges, such as minimizing color usage. To the best of our knowledge, this is the first application of deferral action in the VCP context, integrated with our max-color reward strategy for improved performance. Additionally, our study compares defer and without-defer versions, confirming that deferral action consistently outperforms its counterpart, further validating its effectiveness in the VCP setting.
>
>     **We have summarized these in sections 1 and 2 of the revised version.**
>
>
>
> 2. Thanks for the suggestion. In the revised version, we used 15 minutes per graph to Gurobi in Table 2 so that it can spend sufficient time to find the solution.
>
> 3. Thanks. In our modified version, we have compared our approach with Tabucol in Tables 1 and 2.
>
>
> ## **Responses to Questions**
>
> 1. In this paper, we have discussed and evaluated the VColRL solution and multiple variants of this solution with different rollback strategies. Such a study aims to demonstrate how the variants differ in terms of their performance. Therefore, the proposed solution does not decide when to activate the hard-rollback model versus the soft-rollback model. In a real scenario, one can run both variants and choose the best solution.
>
> 2. We start with 15 colors, as our model is currently designed to output a probability distribution over 15 colors per node, as most graphs, except for DSJC, are 15-colorable. If 15 colors are insufficient or the process is interrupted due to the time limit (in this case, the current color usage count can be less than 15), the uncolored subgraph is extracted. The coloring process restarts on the subgraph, starting from the next color. This continues until all vertices are colored. This is clarified in Section 3.3 of the revised version.
>
> 3) Thanks for the suggestion. We have included the DSJC graphs. The updated results are reflected in Table 2 of the revised version.
>
> [1] Ahn, Sungsoo, Younggyo Seo, and Jinwoo Shin. "Learning what to defer for maximum independent sets." In International conference on machine learning, pp. 134-144. PMLR, 2020.

---

### Note · Authors · 2025-01-23

I have read and agree with the venue's withdrawal policy on behalf of myself and my co-authors.